# Application of Image Analysis for the Identification of Prehistoric Ceramic Production Technologies in the North Caucasus (Russia, Bronze/Iron Age)

**Ki Suk Park [1],\*** , **Ralf Milke [1]**, **Erik Rybacki [2]** and **Sabine Reinhold [3]**

[1]   Institut für Geologische Wissenschaften, Freie Universität Berlin, Malteserstraße 74-100, 12249 Berlin, Germany
[2]   Deutsches GeoForschungsZentrum GFZ, Section 4.2, Telegrafenberg, 14473 Potsdam, Germany
[3]   Deutsches Archäologisches Institut, Im Dol 2-6, Haus II, 14195 Berlin, Germany
\*   Correspondence: ki.suk.park@fu-berlin.de

**Abstract:** The recent advances in microscopy and scanning techniques enabled the image analysis of archaeological objects in a high resolution. From the direct measurements in images, shapes and related parameters of the structural elements of interest can be derived. In this study, image analysis in 2D/3D is applied to archaeological ceramics, in order to obtain clues about the ceramic pastes, firing and shaping techniques. Images were acquired by the polarized light microscope, scanning electron microscopy (SEM) and 3D micro X-ray computed tomography (μ-CT) and segmented using Matlab. 70 ceramic sherds excavated at Ransyrt 1 (Middle-Late Bronze Age) and Kabardinka 2 (late Bronze–early Iron Age), located in in the North Caucasian mountains, Russia, were investigated. The size distribution, circularity and sphericity of sand grains in the ceramics show site specific difference as well as variations within a site. The sphericity, surface area, volume and Euler characteristic of pores show the existence of various pyrometamorphic states between the ceramics and within a ceramic. Using alignments of pores and grains, similar pottery shaping techniques are identified for both sites. These results show that the image analysis of archaeological ceramics can provide detailed information about the prehistoric ceramic production technologies with fast data availability.

**Keywords:** image analysis; ceramic pottery production; North Caucasus; shape parameters; pore topology; 3D μ-CT

## 1. Introduction

In the past twenty years, there has been a rapid advance in microscopy, scanning technologies and image processing [1,2]. Above all, the application of high resolution X-ray computed tomography (CT) clarified the internal structure of fossils, meteorite, textural differences in magmatic, metamorphic and sedimentary rocks and related soils as well as products of sintering procedures [3–8]. This technique is supported by the mass data processing accompanied by the progress in the central processing unit (CPU) and graphics processing unit (GPU) and the data transport system. This allowed the image visualization and segmentation as well as the complex calculation of the geometry of the studied objects such as curvatures of powder grains and interstitial pore topology. Advanced image acquisition and analysis have a huge research potential in archaeology and archaeological sciences. Thanks to the mobility of digital data, researchers can perform the analysis in relatively political and geographical boundary free conditions without the necessity to transport the fragile archaeological objects over long distance and time.

In previous studies, polarized light microscopic images or scanned images of cross thin sections of ceramics were used for the identification of various technical aspects in pottery production, such as ceramic pastes, shaping techniques [9–17]. Alignments and their distributions of grains or pores with specific angles and layout provided clues about the shaping techniques such as coiling, molding or wheel shaping/throwing making use of rotational kinetic energy [16–20]. The 2D image analysis of cross thin sections and 3D analysis using X-ray radiography or 3D micro X-ray computed tomography (3D μ-CT) were employed to characterize the grain size and pore size distribution and to prove specific temper materials in ceramics such as organic materials or heavy minerals [13–15,21,22]. Back scattered electron (BSE) images acquired by scanning electron microscopy (SEM) have provided classical evidence about the temperature dependence of the pore morphology of various ceramics [23,24]. The porosity of the ceramics can be supplemented by the Brunauer–Emmett–Teller (BET) method or mercury intrusion porosimetry (MIP) [25,26].

However, the image analysis of most prehistoric ceramics is still not easy to employ for the study of pottery production technologies, because archaeological ceramics are composed of various heterogeneous mineral phases and chemical compositions, various alteration degrees and firing states within a single sample [27–30]. These properties make it difficult to measure and segment, reconstruct and analyze the region of interest (ROI), the target area of the measurement by the microscopy and 3D-μ CT, from the object in 2D and 3D and this time consuming process is almost impossible to be employed for the whole archaeological ceramics which are found in massive amounts at each archaeological sites. Nevertheless, the research potential of image analysis in archaeological sciences is very large. Various structural elements and components and their shapes and layouts within a ceramic sherd can be directly measured and linked to the various physico–chemical and chemical–mineralogical properties reflecting production site, firing states and forming techniques. The identified techniques will contribute to the reconstruction of prehistoric regional crafts and their technological styles [31–34].

In this study, we tried to learn the extent and possibility of image analysis of highly heterogeneous archaeological ceramics acquired by various analytical instruments which visualize the objects in various scales. Both 2D and 3D images and their parameterization are employed, in order to identify prehistoric ceramic production technologies focusing on the resource gathering, firing degree and shaping techniques. The first and second topics were studied by quantitative evaluation calculating various related parameters and the third one by qualitatively. The image analysis of these characteristics will be performed by Matlab (R2018b), a high performance software which is relatively easily accessible in many research institutes and universities. Ceramic samples were provided from the excavations at Ransyrt 1 and Kabardinka 2 located in the North Caucasus, Russia. During the Bronze and Iron age, both sites yielded ceramics with various contents of Ca, Fe and Si and mineralogical phases including quartz and K-feldspar, plagioclase, mica and alteration products [35]. Firing conditions and associated pyrometamorphic states of the ceramics range from under 400–600 °C to over 1000 °C in both oxidizing and reducing atmosphere. This wide range of the ceramic properties provides a broad set of challenges in the application of this approach.

## 2. Materials and Methods

### 2.1. Archaeological Sites

Ransyrt 1 is located on the plateau with the height of 1850 m above sea level in Karachay–Cherkess Republic of the Russian Federation (43°50′29.7″ N, 42°18′10.3″ E), while Kabardinka 2 lies on the lower plateau with 1400 m a.s.l. in Stavropol Krai of the Russian Federation (43°49′40.9″ N, 42°42′57.4″ E) [36]. According to the local chronology defined by the construction phases and 14C data, Ransyrt 1 is dated to 1800–1500 BC, the Middle Bronze Age (MBA) to the Late Bronze Age (LBA) and Kabardinka 2 to 1600–800 BC, which belong to the LBA and Early Iron Age (EIA) [36]. For comparison to these mountain ceramics, several ceramics found at Levinsadovka (47°10′9.9″ N, 38°30.17″ E) and Saf'janovo (47°15′59.7″ N, 39°26′30.1″ E), which are located in the coast around the Sea of Azov, Mius Peninsula

and on the lower area of the Don river (Russia, late/final Bronze age) were taken [37]. The site at Levinsadovka was occupied by the late Srubnaja culture (LBA) and that at Saf'janovo by Kobjakovo culture (final Bronze age (FBA)) which overlap between 1600–800 BC according to 14C data. The site at Ransyrt 1 was occupied in the middle/late Bronze age and that at Kabardinka 2 in the late Bronze/early Iron age.

*2.2. Materials*

Ceramic samples were selected from 130 sherds which were preselected in 60,000–70,000 pottery sherds found in the settlements in Ransyrt 1 in 2013 and 2015 and at Kabardinka 2 in 2007 and 2008 by optical investigations and transported to Berlin, Germany for the in-depth scientific study. These samples were classified into several groups by chemical and mineralogical analyses with X-ray powder diffraction (XRD), Fourier-transform infrared spectroscopy (FT-IR, absorbance/reflectance), scanning electron microscopy with energy/wavelength-dispersive X-ray spectroscopy (SEM-EDS/WDS) in the parallel study [35] and qualitative estimation of the structures of 2D cross thin sections.

*2.3. Image Acquisition and Reconstruction*

2.3.1. Sand Grains in the Ceramics

In order to characterize sand grains in the ceramics, two-dimensional polarized light micrographs were acquired with a pixel size of $3.27^2$ and $10^2$ $\mu m^2$ for the total area of the cross thin sections. The images by the plane polarized light were used for the analysis and those by the cross polarized light for the comparison and confirmation of the components. We measured 17 samples for Ransyrt 1 and 20 samples for Kabardinka 2.

The 3D image processing of sand grains was performed for selected samples using the 3D μ-CT (nanotom 180NF, GE phoenixIx-ray) with tube voltage and current of 140 kV and 96 μA, respectively, so that the differences between the shape parameters calculated from the 2D images and those from the 3D images can be investigated [38]. A total of 1080 images were taken at angular steps of 0.33 degree and with an acquisition time of 1000 msec/image. The voxel size of $9.49^3$ $\mu m^3$ was determined, in order to compare to the 2D image analysis using $10^2$ $\mu m^2$ per pixel. Considering the heterogeneity of prehistoric ceramics, the ROI was set for the whole area and volume of the sample. The selected magnification of the sample as well as pixel/voxel size were enough to represent the corresponding sample. The acquired images were reconstructed as a volume file using the phoenix datos|x reconstruction software with a beam hardening correction (BHC) factor 8. Each voxel stored values in a 16 bit integer. An edge enhancement filter was applied for the reconstruction.

2.3.2. Pore Topology in the Ceramics

Pore topology was measured by SEM, JEOL JXA 8200 Superprobe with an acceleration voltage of 15 kV for the 2D analysis. The area of $300 \times 300$ $\mu m^2$ was scanned with a pixel size of $1^2$ $\mu m^2$, which allowed to capture various types and sizes of pores in the ceramics. In a 2D matrix, BSE images are presented for each pixel of ROI. The 3D porosity was measured by μ-CT with different tube voltages of 103-129 kV and currents of 70-80 μA according to the sample. The acquisition time for 1080 images ranged from 750 to 1000 msec/image and the voxel size from $1.05^3$ to $3.85^3$ $\mu m^3$ as well. For the image reconstruction, the BHC factor was set to 9-10, in order to minimize artifact effects. 13 samples for Ransyrt 1 and 15 samples for Kabardinka 2 were measured for the 2D image analysis. Three from these samples were selected for the investigation of the 3D porosity (Supplementary Material Table S1).

2.3.3. Formation of the Ceramics

The ceramic formation technique was estimated from the inner structure of 14 samples from Ransyrt 1 and 19 samples from Kabardinka 2 scanned by 3D μ-CT (Supplementary Material Table S2). The three dimensional alignment of sand grains and large pore complexes existing in the inner

structure were taken into considerations for estimating their macroscale formation. Angle, layout and size of those structural elements were investigated in the range of mm to cm, qualitatively. Therefore, the whole ceramic sherd should be within the ROI and this caused a relatively larger voxel size of $9.49^3$–$30.27^3$ $\mu m^3$. The corresponding condition was set to tube voltages of 102–140 kV, currents of 70–103 $\mu$A, acquisition time of 500–1250 msec/image according to the sample. The acquired images were reconstructed by the phoenix datos|x reconstruction software with a BHC factor 8, voxel values in a 16 bit integer and the edge enhancement filter.

### 2.4. Image Segmentation and Parameterization

For the image segmentation and parameterization, the acquired 2D and 3D images were calculated with Matlab (Supplementary Material Table S3). For the quantification of sand grain shapes in 2D, the plane polarized micrographs were converted to the gray scales to reduce the color variances and to raise the accuracy in segmentation of heterogeneous phases. Various image filters and retouching the images were used as well, in order to achieve precise and fast segmentation. After segmentation, numeric properties of shapes and pores in 2D were calculated by the CPU using eight-connectivity/neighborhood of pixels (Supplementary Material Figures S1 and S2), while the calculations of 3D objects were performed by CPU and GPU using six- and 26-connectivity/neighborhood of voxels (Supplementary Material Figure S3). The connectivity of pixels and voxels determines a same region by counting adjacent unit cells. Eight-connectivity defines the 2D area sharing same vertices. In 3D image processing, the same region calculated by six-connectivity shares identical faces with the adjacent cells, while that by 26-connectivity shares edges, faces and vertices. The details of each parameter will be discussed in the following.

## 3. Results and Discussion

### 3.1. Characterization of Ceramic Pastes by Shape Parameters of Sand Grains

Most ceramic pastes in the ancient material production contain sand grains. Especially, coarse sand grains composed of lithoclasts bigger than 250 $\mu$m reflect the environment where they are originated well and keep their form during the firing of the ceramics by the grain size effect. Moreover, they can be taken easily from the cross thin sections in 2D which are prepared in many ceramic studies. According to the parallel mineralogical and chemical study, grain forming dominant minerals are presented in Figure 1 [35]. Because most sands in a ceramic object are composed of various mineralogical phases in various textures and the shapes of these poly-mineral grains were changed during the sedimentation process, shape parameters of the sand grains should be calculated for each grain and not for each single mineral phase. In the samples, lithoclasts and sedimentary aggregates composed of similar mineral phases exist in various sizes, indicating a possibility that the resource were gathered in the sediments accumulated by the natural depositional process.

In this study, we calculated the following shape parameters: maximum length, circularity, sphericity 1 and sphericity 2 of an idealized ellipse that has the same normalized second central moments as the segmented object. They were defined as [39–41]:

$$\text{Circularity}: \frac{4\pi A}{P^2}, \tag{1}$$

$$\text{Sphericity 1 (elongation)}: \frac{D_{F,max}}{D_{F,min}} \tag{2}$$

$$\text{Sphericity 2 (elongation)}: \frac{a}{b}, \tag{3}$$

where $A$ and $P$ mean area and perimeter, $D_F$ is Feret diameter, $a$ and $b$ are major axis minor axis of an idealized ellipse of the grains, accordingly.

Figure 1 illustrates different size distribution, circularity, sphericity 1 of the coarse sands according to the archaeological sites and the dominant mineral groups in the sand grains in the ceramics. The average and standard deviation of the shape parameters of individual grains in each sample were calculated according to the site at Ransyrt 1 (Figure 1a), Kabardinka 2 (Figure 1b) and Levinsadovka–Saf'janovo (Figure 1c). In general, the ceramics found at Ransyrt 1 contained bigger sized grains with a wider size distribution and less circular shapes than the other sites. The ceramics excavated at Levinsadovka and Saf'janovo have more rounded and finer sands. Despite the high distribution of the mean sand size, there is a tendency that the size and angularity of coarse sands in the ceramic pastes decrease from higher mountains through middle plateau to the alluvial sea coast.

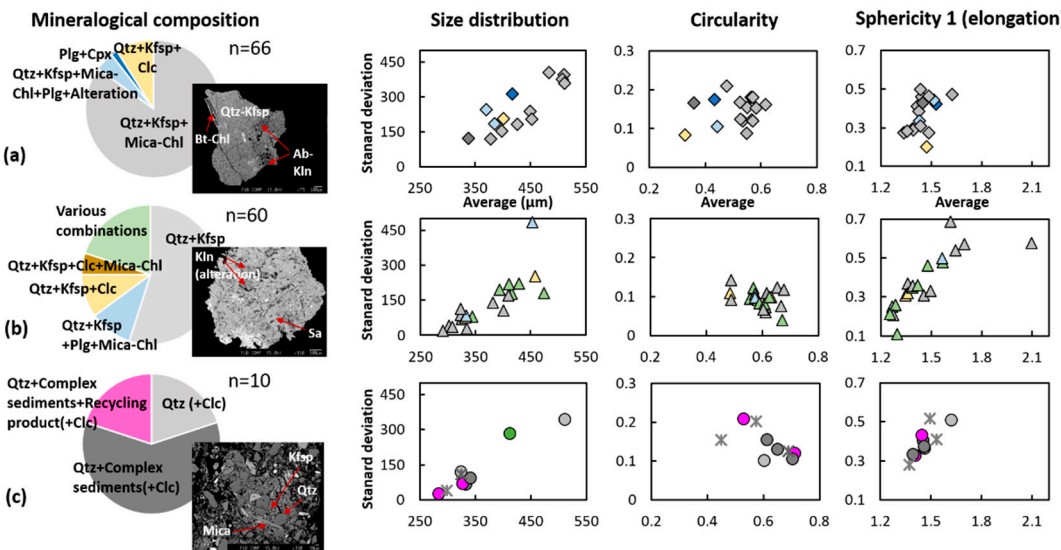

**Figure 1.** Mineralogical composition of dominant grain groups of the samples for the sites, (**a**) Ransyrt 1, (**b**) Kabardinka 2, (**c**) Levinsadovka–Saf'janovo with the back scattered electron (BSE) images of the representative grains [35] is compared to the size distribution, circularity and sphericity 1 of coarse sand grains (>250μm, lithoclasts) present in the selected samples (Clc: calcite, Cpx: clinopyroxene, Kfsp: K-feldspar, Sa: sanidine, Mica–Chl: mica–chlorite mixed layers, Bt: biotite, Ol: olivine, Plg: plagioclase, Qtz: quartz, Kln: kaolinite). Each mineralogical grain groups presented in different colors (Qtz + Kfsp + Mica – Chl/Qtz + Kfsp/Qtz (+Clc): light grey, Qtz + Kfsp + Plg + Mica Chl + Alteration/Qtz + Kfsp + Plg + Mica – Chl: light blue, Plg + Cpx: blue, Qtz + Kfsp + Clc: light yellow, Qtz + Complex sediments (+Clc): dark grey, various combinations: light green).

The highest elongation degree of the grains of a sample group representing for Ransyrt 1 is bigger than those in the ceramics from the other sites. At the same time, the shape parameters of the sherds are varying within each site, where two or three groups can be classified within the ceramics from the same site. It is possible that the distribution of the mean value of the calculated shape parameters becomes smaller, if the measurement area is bigger or the several sherds originated from a single pottery are analyzed. Concerning the relation between the mineralogical combination of the coarse sands and their shape parameters, the Ransyrt 1 samples show this correlation in the circularity very slightly, while this is found in the size distribution and sphericity 1 for the samples from Kabardinka 2. However, those parameters derived from the 2D image processing would represent a relative value regarding the three dimensional alignment of grains in the ceramics. Figure 2 shows the comparison between the grain size distribution, sphericity 1 and sphericity 2 calculated with the 2D and 3D image analysis of an example. In comparison to the 2D data, the elongation degree of the coarse sands from the 3D images is higher and this attests to the fact that elongation vertical to the observed sample

surface is detected in 3D but not in 2D. The alignment of sands can therefore influence the results of 2D image analysis.

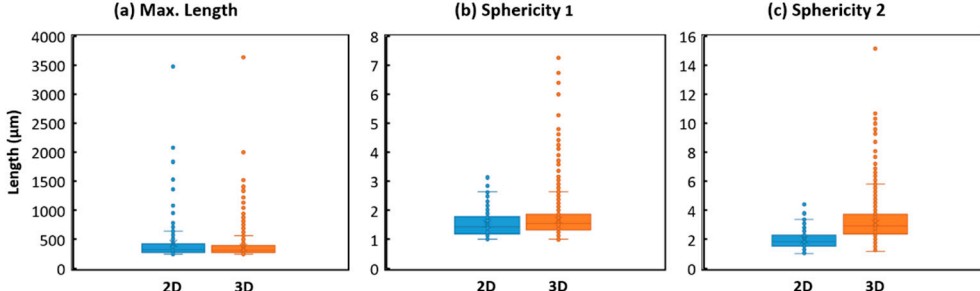

**Figure 2.** (**a**) Grain size distribution (max. length), (**b**) sphericity 1 and (**c**) sphericity 2 derived from 2D (blue) and 3D (orange) image analysis of the coarse sand grains (>250 μm) in the ceramic sherd (Ran1_514_2). Pixel size of $10^2$ μm$^2$ and grain numbers/A (area) = 1.501 ea/mm$^2$ for the 2D images. Voxel size of $9.49^3$ μm$^3$ and sand grain numbers/V (volume) = 12.182 ea/mm$^3$.

Although the ceramic pastes could be prepared by the addition of various sediments and removal of specific components by the potters, the grain size distribution of the coarse sand grains in the ceramics reflect the soil development of each site concerning the sedimentation process from the mountains to the alluvial zones [35,42]. The circularity and sphericity of coarse sand grains show site specific geological setting as well. Because Ransyrt 1 on the higher plateau has less soil than Kabardinka 2 on the lower plateau and the sites around the alluvial zone, the potters would have gathered resource for the preparation of ceramic pastes around the settlement sites.

However, every archaeological site has various groups of samples according to each shape parameters. This means that the potters used different strategy in the gathering, preparation or control of ceramic pastes. Grain-like cavities proved in a few samples by the 3D scanning are possibly produced by the thermal decomposition of calcites during firing (Figure 3) [43]. This informs us that calcite grains were used intensively in a few ceramics.

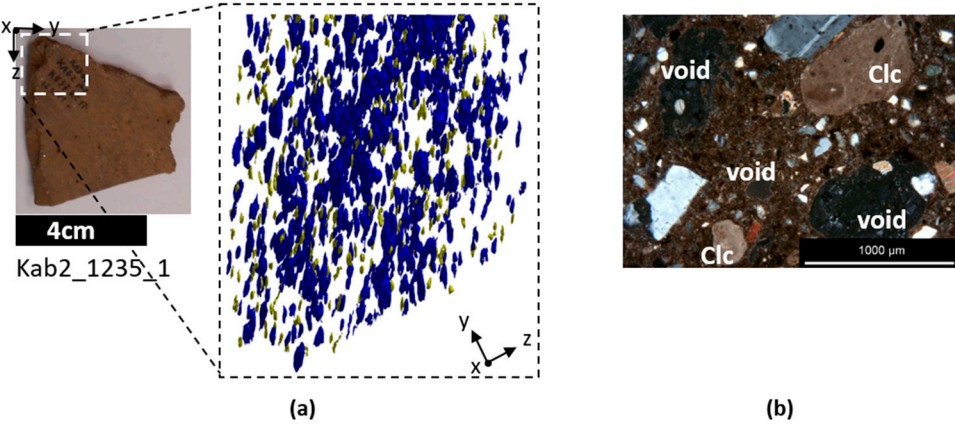

**Figure 3.** (**a**) Original ceramic sherd and segmented grain like cavities, calcite ghosts in the sample (blue and yellow) (voxel size: $25.00^3$ μm$^3$). (**b**) A cross polarized light micrograph of the cross thin section shows coexisting calcite grains and closed type of voids in different sizes (Clc: calcite).

## 3.2. Pore Topology and Pyrometamorphic Degree

Pyrometamorphism of archaeological ceramics indicating firing degree and conditions has been estimated by chemical and mineralogical changes and clay sintering by various analytical instruments such as XRD, SEM and FT-IR [24,44]. Pore topology is one of the key variables related to the

pyrometamorphic degree of the ceramics, because firing temperatures and atmospheres change pore shapes and orientations caused by thermal expansion and sintering of clay minerals [45–47]. If particles with identical shape and composition are sintered, the interstitial topology changes in three steps: (i) the contact between neighboring particles (ii) the interconnected channels with cylindrical pores (iii) the formation of closed pores [45,46]. Clay sintering has a similar process as well [47]. The initial stage is the formation of sharply concave necks between the individual particles, while the intermediate stage forms a 3D interpenetrating network of solid particles and continuous-, channel-like pores with high curvatures. The final stage of sintered clay shows a huge decrease in porosity with isolated and closed forms. Because the prehistoric ceramics contain various scales and anisotropic shapes, the densification process and the coarsening process in the sintering will occur as a mixed process [47]. Despite of the difficulties in the morphological description of this heterogeneous system, the morphological changes in the clay matrix were described by several simplified steps during ceramic firing in a previous study [48]: (i) drying and shrinking of the clay paste (ii) dehydration in the low temperature firing interval between 100 and 200 °C, creating interstitial pores (iii) continuous increase in porosity by chemical reactions such as dehydroxylation of clay minerals between 400 and 800 °C (iv) liquid phase sintering forming interconnected pores of irregular shape and partial melts (v) micro-fabric formation ranging from non-vitrified to completely vitrified final state. In the parallel study, the selected ceramic samples were ordered according to the different micro-morphology and mineralogical changes, which range from low firing through thermal expansion and dehydroxylation to the early melting stage [35]. In general, the fabrics of the samples could be classified into four types in terms of micro-pores and colors of ceramic matrix as well: (1) elongated open pores with the matrix in dark brown/brown/gray/black; (2) closed end-elongated pores with the matrix in light brown/reddish brown/dark red; (3) open pores/closed end-elongated/closed globular pores with the matrix in light gray/light red; (4) closed globular pores in orange/dark gray.

The morphological changes led by clay sintering can be described by the shape parameters such as pore size/area/volume distribution, sphericity 1 and sphericity 2. Additionally, the complex pore geometry can be expressed by the pore topology function of the archaeological ceramics such as Minkowski functionals [49,50]. Especially, the Euler characteristic ($\chi$) provides an unbiased estimation of pore topology, applied to a 3D cutout of arbitrary shape and volume [50]. It was calculated from 2D and 3D images using the following formula:

$$\chi_{2D} = n(objects) - n(pores), \tag{4}$$

$$\chi_{3D} = n(objects) - n(tunnels) + n(cavities), \tag{5}$$

where $\chi_{2D}$ and $\chi_{3D}$ represent the Euler characteristic in 2D and 3D and n(objects), n(tunnels) and n(cavities) are the total numbers of objects, tunnels and cavities, respectively. However, due to the complex structure of ceramics, these elements for the Euler characteristic in 2D and 3D are difficult to count directly. For the application of this characteristic in the heterogeneous ceramic structure, the Equation (5) is replaced by the Euler–Poincaré formula based on a cubic form of the voxel which can be acquired directly from the reconstructed images [51,52]:

$$\chi_{3D} = n(vertices) - n(edges) + n(surfaces) - n(volumes), \tag{6}$$

where n(vertices), n(edges) and n(surfaces), n(volumes) mean the total number of vertices, edges, surfaces and volumes of the measured ceramic objects. Due to the very heterogeneous shapes existing in the samples, this calculation used the six-connectivity of voxels defining the areas of pores and ceramic matrix. The orientation of the pores which were mainly decided by the forming and shaping actions of the potters can provide additional evidence regarding the sintering stage. In 2D image analysis, the angle between the x-axis, the cross profile of the ceramic thin section in this study, and the major axis of the ellipse that has the same second-moments as the pore region, referred as θ, was used.

It ranges from –90 degrees to +90 degrees. In 3D data processing, Euler angles were taken for the x-(φ), y- (θ) and z-axis (ψ) based on the right-hand rule [53,54]. In order to normalize porosity acquired from the different areas and volumes of ROI, Euler characteristic is divided by the corresponding area of the object for 2D and volume for 3D. In 2D images for the area of $300 \times 300$ μm$^2$, average and standard deviation of sphericity 2 of individual pores become smaller due to clay sintering, while Euler characteristic suddenly decreases just before the beginning of the liquid sintering and after that it increases again (Figure 4). Due to the different grain size distribution, the numeric thresholds suggested by this Euler characteristic are formed differently according to the archeological sites.

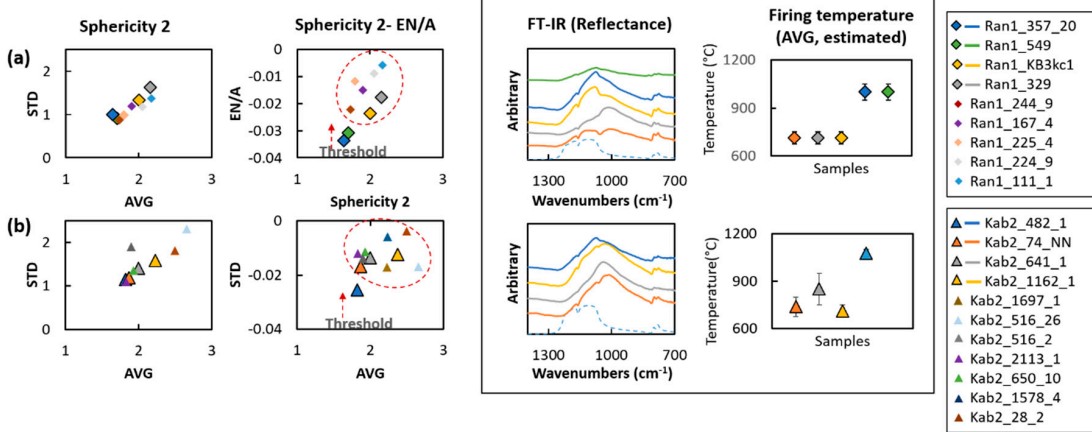

**Figure 4.** Average and standard deviation of sphericity 2 of pores and comparison between Euler number per unit area of the pores (A = $300 \times 300$ μm$^2$) in average and sphericity of pores in average. Calculated from the pores (2D) present in cross thin sections of the samples from (**a**) Ransyrt 1, (**b**) Kabardinka 2. These parameters of four samples for each site (bigger maker with a boundary border) were compared to the reflectance Fourier-transform infrared spectroscopy (FT-IR) (aperture size: $70 \times 70$ μm$^2$, reference: silver mirror) [35].

The pyrometamorphic degree of each selected sample measured by the reflectance FTIR ($70 \times 70$ μm$^2$, silver mirror) and corresponding deformation degree estimated by multi-proxy methods support this parametric change [35]. However, this property has a dependency on the grain size and shape, thus ceramics found at Ransyrt 1 and Kabardinka 2 show a different distribution of Euler characteristic before the sintering. Moreover, the firing state within a single sample is heterogeneous, so that the average temperature of the whole ceramic sherd does not always coincide with the local firing state [35]. Despite of this high degree in the heterogeneity, these parameters provide thresholds of the sintering degree of the ceramic pastes, especially the beginning of the liquid sintering phase. The Euler characteristic calculated from each area within a same sample provides access to a spatial distribution that indicates the localization of the pyrometamorphic degree (Figure 5).

In Figure 6, the 2D images and 3D segmented pores of the same samples are compared to each other. According to the increase in the degree of sintering, the pore volumes decrease in size. These samples contain a relatively similar percentage of sand size distribution (Figure 7). The degree in orientation of the closed pores, θ decreases as well. The sample with the highest pyrometamorphic state (Ran1_549) shows the highest variations of the localized Euler characteristics. Interestingly, the Euler characteristic calculated from the 2D and 3D images does not show a linear relation and rather is related to the higher interconnectivity of the pores and grains in the 3D images. This reflects the faster disappearance of pore connectivity in the x–y-plane of the wall fragment than in z-direction. These changes of the pore shapes to an ellipsoidal form is controlled by vacancies diffusion ruling pore morphology during sintering, because the large pores created with different curvatures favor this shape according to the increasing sintering degree [55]. Figure 8 shows a tendency of the closed pore

networks in 3D volume to become more elongated in accordance with metamorphic degree, while their surface area and volumes decreases. The orientation degree, ψ, of the individual open and closed pores in the ceramics decreases according to the increasing pyrometamorphic degree. This means a rotation around the z-axis in the clockwise direction according to the increasing firing temperature.

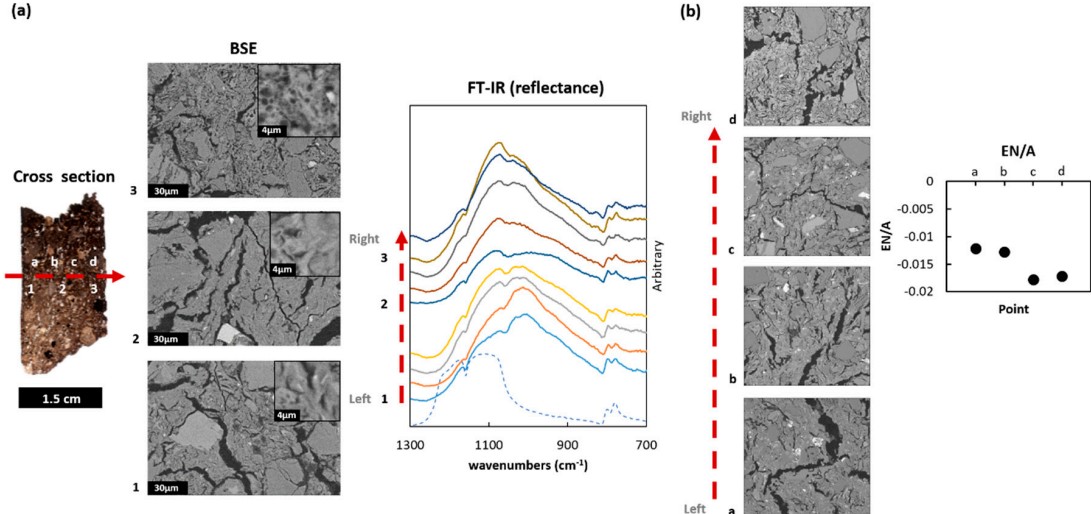

**Figure 5.** (**a**) Cross section profile of the reflectance FT-IR (aperture size: $70 \times 70$ μm$^2$, reference: silver mirror, dashed line in blue: quartz) [35] and corresponding BSE images for the left side (1), middle (2) and right (3) of the sample (Ran1_167_4). (**b**) Cross section profile of the Euler characteristic per unit area and related BSE images from the left side (**a**) through left middle (**b**) and right middle (**c**) to the right side (**d**) of the same sample in a same direction. Calculated from the two-dimensional area of $300 \times 300$ μm$^2$ on the cross thin section.

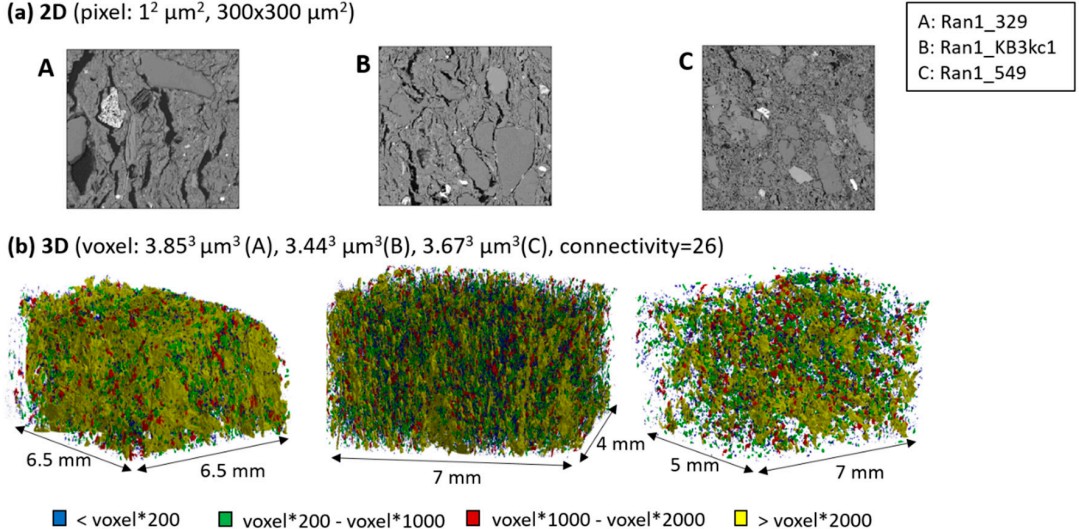

**Figure 6.** Comparison of the pore topology between (**a**) 2D and (**b**) 3D images of three samples varying in the degree of pyrometamorphic degree/sintering [35]. Estimated firing temperature for A: 700–850 °C, B: 700–850 °C and C: 950–1050 °C. The segmented pores show the difference of the firing degree between A and B more clearly.

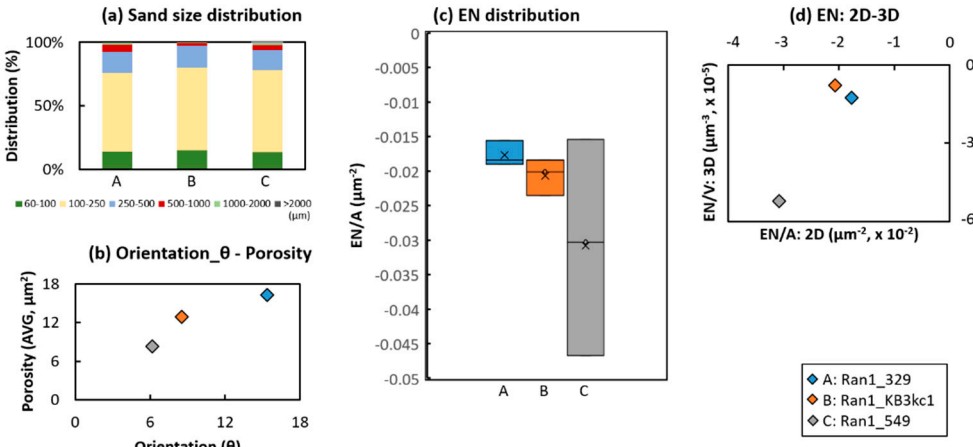

**Figure 7.** Comparison of sand size distribution calculated by (**a**) 2D images and (**b**) orientation of two-dimensional pores, (**c**) spatial distribution of the localized Euler number within a two dimensional unit area and (**d**) Euler number (average) derived from 2D images and that from 3D images (connectivity = 6). Samples (A, B, C) from the Figure 5.

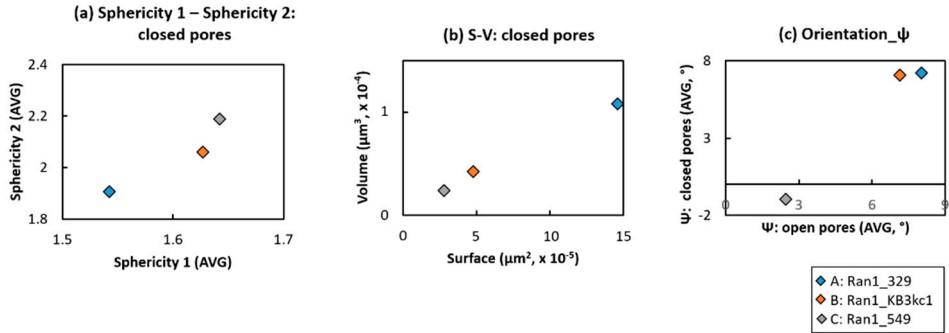

**Figure 8.** Relationship between the average of (**a**) sphericity 1 and sphericity 2 of closed pores and (**b**) surface area and volume relation of the closed pores and (**c**) the comparison of orientation degree ψ and the closed and open pores in average. All parameters derived from 3D scanning with the connectivity 26. Samples (A, B, C) from the Figure 5.

These results show that the segmented pores in 2D and 3D and their parametrization according to the sintering stages are useful as an additional indicator supporting the estimation of the degree of pyrometamorphism of the archaeological ceramics. Although the parameter values from the 2D and 3D image analysis provide results slightly different from each other, which is probably caused by the different unit cell size and counting of the objects, they can both be employed for the classification of the archaeological ceramics according to the firing degree. The Euler characteristic provides the numeric thresholds of the sintering stages of each sample. Especially, the transition into the liquid sintering of the ceramic matrix can be distinguished by the huge decrease in this property, which is important for identifying high firing ceramics at the site. Despite the complex pore structure caused by the heterogeneous composition in the ceramic paste, ceramics with similar degree of heterogeneity can be analyzed by this parameter describing the pore topology and related sintering.

In comparison to the Ca-poor ceramics, the samples containing considerable amounts of Ca-carbonates contain additional pore type caused by the thermal dissociation carbonate grains [35]. This type of porosity is mainly controlled by a combination of the firing degree and the size of carbonate grains. Thus, the shape parameters of the Ca-carbonate rich ceramics cannot be compared to each other as well as the Ca-poor samples.

### 3.3. Shaping Techniques: Inner Structure

Despite of various reasons and factors generated by the potters during the production or natural resources, use of the ceramics or weathering process, 3D alignments of macro-pores and coarse sand grains in the samples can provide a possibility to assume the formations of the inner structure of ceramic bottom and wall respectively. Normally, the identification and categorization of such formations can be easily misled by the optical surface observation due to the weathering process and can be limited in size and scale by the investigation using destructive 2D cross sections of ceramics. Therefore, this case study focused on the images of structural elements acquired by the 3D μ-CT in a range from mm to cm.

Figure 9 shows a classification of the inner formation of the bottom and wall part representative for Ransyrt 1. In the bottom-wall fragments, four types of alignments were identified: (1) most pores existing in the bottom plate are not connected to those in the wall. Only one long pore complex continues from the bottom to the wall. In the bottom plate, the pores are aligned parallel in the horizontal direction (Figure 9(a1)). (2) The pores from the bottom link to the wall. The pores in the bottom are lying parallel to each other (Figure 9(a2)). (3) The pores in the bottom and wall do not connect to each other. The pores of the wall are vertically aligned starting from the side of the bottom part (Figure 9(a3)). (4) The bottom plate consists of two layers (Figure 9(a4)).

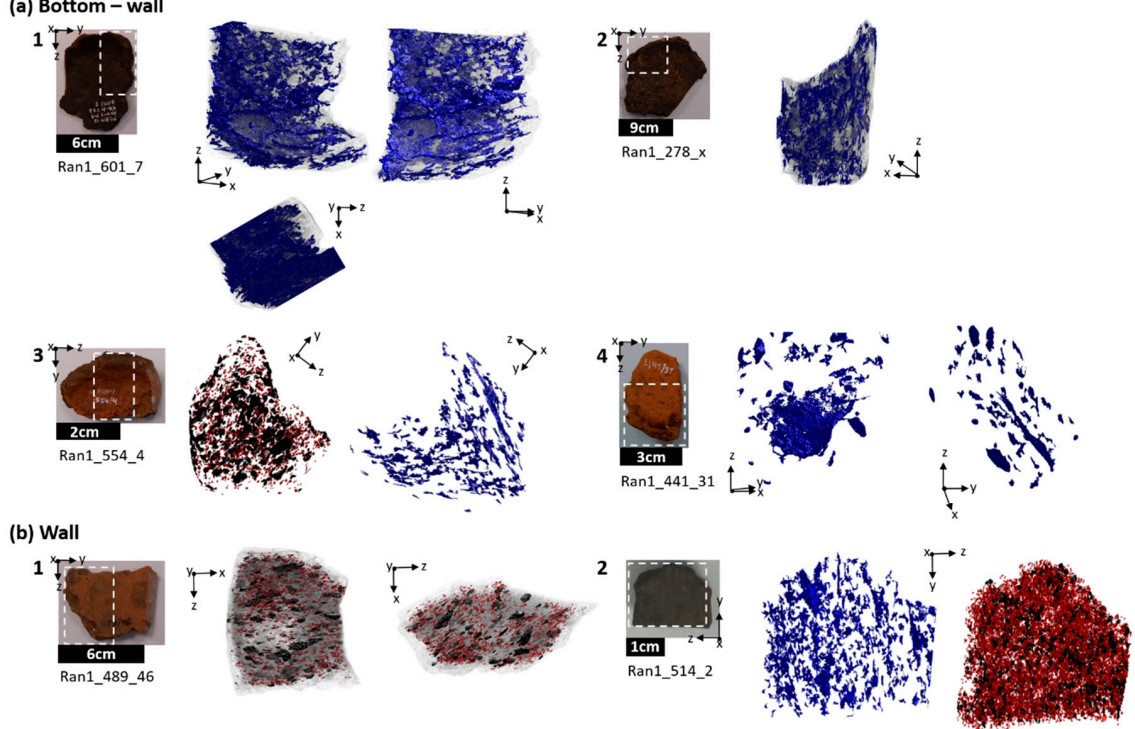

**Figure 9.** Photos of original ceramic sherds and segmented images of the large pore complex (blue) and sand grains (red and black) of the ceramics discovered at Ransyrt 1 (connectivity = 26). Region of interest (ROI) in the box of the white dashed line. (**a**) Bottom-wall fragments: 1) separate formation of the bottom and wall part (wall on the bottom), oriented parallel pressure/stress from the surface (voxel size: $29.05^3$ μm$^3$) 2) continuous formation of the bottom and wall part (voxel size: $29.63^3$ μm$^3$). 3. separate formation of the bottom and wall part (wall next to the side of the bottom) (voxel size: $24.31^3$ μm$^3$). 4. two layers for the bottom building (voxel size: $28.21^3$ μm$^3$). (**b**) Wall fragments: 1) mixture of regular and irregular alignment of sand grains (voxel size: $19.33^3$ μm$^3$). 2) Vertical orientation of the large pores in comparison to the alignment of the sand grains (voxel size: $9.49^3$ μm$^3$).

The wall fragments can be categorized into two groups in general: (1) irregular alignment of pores and sand grains, some grains showing specific orientation (Figure 9(b1)). (2) Most grains and large pores are oriented to the vertical direction (Figure 9(b2)). In general, the alignment of the pores and grains indicates the direction of the uniaxial compression and bending force applied by the potter in the walls during the modelling phase, where irregular directions of the pores and grains can be created easily by the hand shaping. Most inner structures of the selected samples were formed by these internal forces would be created by the pressing by potters with various intensities.

According to the combination of these identified alignments of pore complex and sand grains, representative shaping techniques for the bottom and wall fragments can be supposed accordingly: (1) The wall and bottom were formed into a separate part and the wall is added on the bottom later. Basically, both parts were shaped by pressing, to the horizontal direction for the bottom part. (2) The wall and bottom part were built together. The bottom was flattened by pressing from inward out. (3) The wall was pulled out or pinched slightly to the vertical direction and was added to the side of the bottom plate. (4) The wall and bottom were built separately and bound together later. (5) The bottom plate of the ceramic pottery was formed by two layers for the bottom plate. It is probable that the potters shaped the walls usually by hand. At the end, they pressed the wall, probably with the fingers or palms, for the compactness and finishing and bent. Regarding the ratio of coarse sand grains in the paste identified in the Section 3.1 of this study, tensile stress created by pulling would not be employed for the shaping of a pottery, because the paste would not be so flexible. The combination of several parts composed of coarse grains with similar orientation could be interpreted that the potters added the wall fragments side by side or stacked them from the bottom to the upside part such as coiling (Figure 9(b1)).

In this way, the ceramics found at Kabardinka 2 can be categorized into a few groups of formation as well (Figure 10). Three types were identified for the main layout of the bottom—wall part: (1) the large pores connect both parts (Figure 10(a1)). (2) The pores from the bottom and wall part are not connected. The pores in the bottom show a rounded form, while the pores in the wall are elongated in the vertical direction (Figure 10(a2)). (3) The pores in the bottom part are not linked to the wall part. The bottom part contains pores aligned in parallel. Some of them on the surface are very sharp cuts (Figure 10(a3)).

The wall fragments excavated at this site contain clearer traces of their making than those found at Ransyrt 1 due to the finer grain size distribution of the ceramic pastes: (1) The pores are aligned in the vertical direction (Figure 10(b1)). (2) The pores are aligned in the vertical direction from the side view, while from the front they are oriented in irregular direction (Figure 10(b2)).

The similar alignment of grains and pores indicate that the potters at Kabardinka 2 shaped the bottom and wall part of the ceramics using the same compression and bending force in similar ways: (1) Both parts were built separately and pressed together. The bottom was flattened leaving round traces and the wall was formed to a vertical direction. Some of the bottom plate was done by a tool with a sharp edge. (2) The wall part was drawn, pinched or bent from the bottom part. (3) Both parts were formed separately and the bottom was flattened by pressing into the horizontal direction.

There are no ceramic sherds containing any sign for the rotational kinetic energy which would let the grains in the pastes aligned in the second direction with 5–60° and which would prove the application of the wheel-throwing/shaping [16–18,21]. Thus, the ceramics from both sites were basically made by hand and shaped by the formation of the whole objects or by the modular slab building.

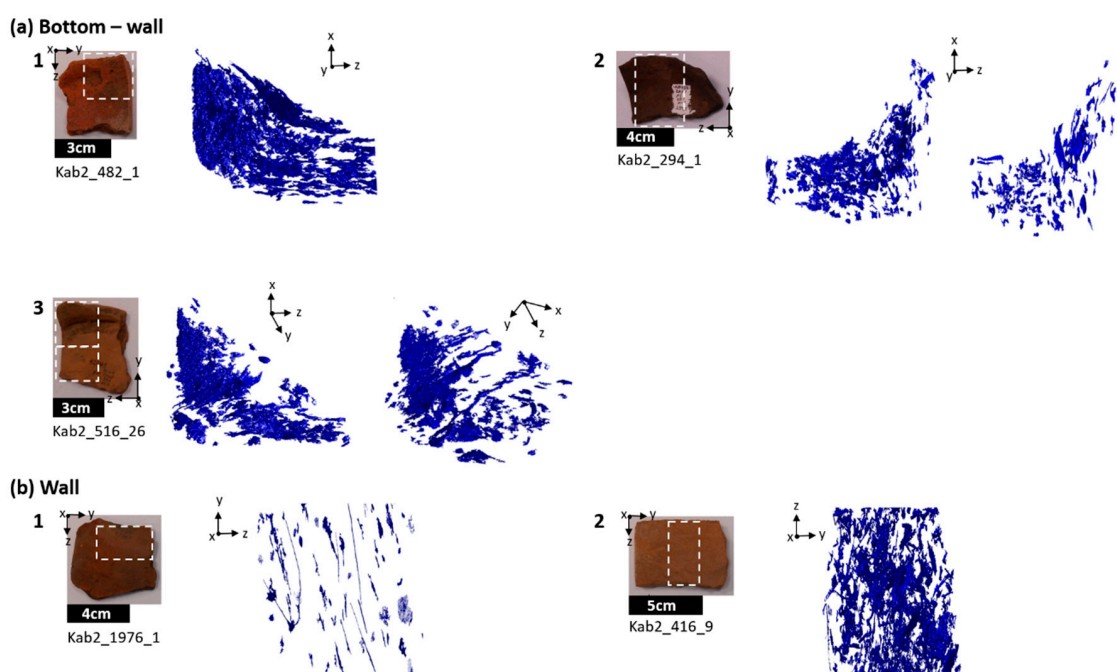

**Figure 10.** Photos of original ceramic sherds and segmented images of the large pore complex (blue) and sand grains (red and black) of the ceramics discovered at Kabardinka 2 (connectivity = 26). ROI is in the box with the white dashed line. (**a**) Bottom-wall fragments: 1) continuous formation of the bottom and wall part (voxel size: $20.00^3$ μm³). 2) Separate formation of the bottom and wall part (wall next to the side of the bottom) (voxel size: $25.00^3$ μm³). 3. separate formation of the bottom and wall part, oriented parallel pressure/stress from the surface (voxel size: $17.31^3$ μm³). (**b**) Wall fragments: 1) alignment of the pores to the vertical direction (voxel size: $27.04^3$ μm³). 2) Rough alignment of the pores to the vertical direction with the random direction from the front view (voxel size: $25.00^3$ μm³).

## 4. Conclusions

Direct 2D and 3D measurements of structural elements of the original ceramic samples and their parametrization provided qualitative and quantitative information for the various topics in the archaeological ceramic studies such as resource gathering, pyrotechnology and shaping techniques of inner structure in macro-scale. Despite of heterogeneous structures and diverse mineralogical and chemical composition, the samples showed inter- and intra-site-specific shape and morphological parameters, as long as appropriate scales of measurement areas/volumes, spatial resolutions, shape parameters and connectivity in the unit cell for the defining a same region are selected according to the purpose. However, shape parameters acquired from 2D and 3D image processing of the same samples cannot be compared directly due to the limitations of 2D geometry for 3D objects. This application for deriving morphological factors in heterogeneous composite materials provides a fundamental approach for studying the details of the sintering kinetics of the archaeological ceramics which often requires the transformation of the reconstructed images into the 3D triangle meshes for the more complicated memory consuming calculations. The 3D alignment of pores and grains on a macroscale could suggest the categorization of the whole/modular formation of slabs at the sites, despite of the wide ranges of grain size distribution in the objects, especially for the ceramics found at Ransyrt 1. Moreover, these whole applications to the original samples provided fundamental information for the experimental studies, so that the results from the original objects and their replicas can be compared to each other. The high applicability of the digital data and image processing with the flexible and easily accessible software will raise the research potential of the 2D and 3D image analysis in the archaeological science.

**Supplementary Materials:** The following are available online at http://www.mdpi.com/2571-9408/2/3/143/s1, Figure S1: (a) Polarized light microscopic images (plane polarized light) of the cross section; (b) segmented image of sand grains (white), Figure S2: Visualization of the 2D matrix according to the intensity of (a) BSE and (b) Carbon and segmented image of pores from the matrix (measurement area = $300 \times 300$ µm$^2$), Figure S3: (a) Original ceramic sherd and its ROI in the box of the black dashed line; (b) Isosurface of the sample with Isocaps of the exposed cross section; (c) Segmentation of sand/silt grains and pores; (d) Segmentation of sand grains and pores with the specific volume size, Table S1: Measurement condition of the 3D µ-CT for the pore topology of three samples, Table S2: Measurement condition of the 3D µ-CT for the alignment of the pore complex and sand grains, Table S3: Examples of main functions used for the image analysis by Matlab in this study.

**Author Contributions:** K.S.P. designed the experiments, performed the measurements and image processing, analyzed the data and wrote up the manuscript; R.M. conceived of the study and controlled the measurements for the 2D images and reviewed the manuscript; E.R. controlled the measurements by the 3D µ-CT and data and reviewed manuscript; S.R. provided the archaeological data and contexts. All authors gave final approval for publication and agree to be held accountable for the work performed therein.

**Funding:** This research was funded by the state of Berlin's Elsa–Neumann scholarship (NaFöG). The publication of this article was funded by Freie Universität Berlin.

**Acknowledgments:** The authors wish to thank D.S. Korobov from the Institute of Archaeology, Russian Academy of Sciences and A.B. Belinskij from GUP Nasledie, cultural heritage organization in the Stavropol Region, Russia for the support for this field work. We are also grateful to D.V. Zhuravlev, the State Historical Museum, Moscow for the export of the samples for scientific analysis in Germany. We acknowledge L. van Hoof, Freie Universität Berlin and O. Dally, Deutsches Archäologisches Institut, Rom, for the ceramic samples from the Taganrog archaeological project. For the access to the polarized light microscopy with a low magnification, we would like to thank P. Groß, Institut für Geologische Wissenschaften, Freie Universität Berlin.

**Conflicts of Interest:** The authors declare no conflict of interest.

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
