# Peer review of "Application of Image Analysis for the Identification of Prehistoric Ceramic Production Technologies in the North Caucasus (Russia, Bronze/Iron Age)"

_heritage, doi:10.3390/heritage2030143_

Round 1
Reviewer 1 Report
The paper provides an extensive experimentation using a fan of different techniques and analysis methods to interpret the provenance, usage of materials, manufacturing style and shaping techniques in prehistoric ceramic shards, based on the shape properties and structure of sand grains and pores in the ceramic paste.
The study deploys a lot of different investigation techniques to acquire image and volumetric representation of samples, and applies various analysis methods to investigate different aspects: the characterization of sand grains, the porosity and pore topology (in 2d and 3D).
There is so much information that is a bit difficult to digest, especially concerning section 2. I suggest to structure this section making an effort to distinguish what is done for sand grains, in 2D and 3D, and what for pores (2D and 3D) in different subsections. This would help the reader, I think. Also, some general considerations could be anticipated, like the first paragraph os section 3.1. I mean, better explain in advance what macroscopic features halp in interpreting the shard and then go at microscoping level to measure such features.
Please define what you mean by ROI - I assume region of interest - this means a small shard portion where microscopic data are acquired?
I don't think I understand this sentence in section 3.1: "Average and standard deviation... were derived from individual grains in the image responsible for each sample"... could you rephrase please?
In equation 6, we are refering to the volumetric grid of voxel, right? so we could specify that entities in the Euler characteristics are vertices, edges, faces and cubes, is that correct?
I think a strong point of this paper is to use different approaches to measure and qualitatively evaluate the inner ceramic structure. Very nice experiments, well detailed, support interesting findings. I wonder is more adaptive 3D representations (suche as tetrahedral or hexahedral meshes) could approximate better pores and sand grains and improve accuracy.
Author Response
Thank you very much for the comments about the research potential and your suggestions for the improvement. As you suggested, we corrected some points in the manuscript.
1. We restructured the section 2 according to the archaeological site/materials/image acquisition/image segmentation and parameterization.
2. The ROI, Region Of Interest, is used as the area what is/will be measured in microscopy/CT, as you wrote. Because the ROI is to set the measurement conditions, especially in CT, we employed this term in the manuscript. We added this definition to the manuscript.
3. We rephrased the sentence more clearly.
4. Yes. The Euler Characteristic in 3D can be estimated by the basic definition (5) and applied formula (6) which is more applicable in the heterogeneous structures. We rephrased the sentences related to the explanations for the eq. (5) and (6), so that the readers can see more clearly, why and how the basic definition (5) is replaced by the eq. (6).
5. We tried the tetrahedral and triangle meshes for the presentation of the segmented pores and grains. The details of the shapes are better presented in the direct segmentation of the reconstructed images, however it required huge RAM for the further calculation such as curvatures. The use of specific meshes provided more fast and memory saving calculations and enabled to use triangulation by Matlab for the further properties, which was not presented in this study, because the topic was different. Short comment about the possibility is added to the conclusion part.
We appreciate your comments again.
Reviewer 2 Report
I know the problem of applying imaging techniques to identify technological features in prehistoric ceramics and although I am aware of the impossibility of applying it to all the ceramics found on archaeological sites, I find the demonstration of how the technique, even with different intrinsic limits, is really useful. There is only few point to reconsider.
Line 87: "Mius peninsular" probably it is "Peninsula"?
Caption of Figure 1: ceck the spacing beetween the words
Figure 3: can you enlarge the pic in order to make more visible the calcite ghosts?
line 262: Minkowski distance function?
Line 265-266: why do you mention n(tunnels) and n(cavities), where they are in the formula?
Author Response
Thank you very much for the comments about the research potential despite of the general problems in the ceramic studies. We checked your suggestions in the manuscript.
1. We corrected the spell. Sorry about the basic mistake.
2. We checked the caption of the figure 1,9,10 and corresponding numbering in the text.
3. We enlarged the figure 3, so that the reader can see the structure more clearly.
4. We rephrased the sentence related to line 262, that the elements of n(objects), n(tunnels) and n(cavities) of the Euler characteristics in 2D and 3D are not easy to get directly from the heterogeneous complex structures.
5. We rephrased the sentence, why the formula (5) based on the definition of Euler Characteristic (counting objects, tunnels and cavities) is reformulated by the Euler-Poincare formula, so that the reader can understand the change from the formula with basic definition to the formula in the applied form.
We appreciate your comments again.
Reviewer 3 Report
Please check the enclosed file.

Author Response
Thank you very much for the comments about the research potential and your suggestion for the improvement. We think that BET is a very precise method for the measuring porosity which can be compared to the results from the image analysis. But due to the limit in the measuring closed pores in the ceramics, we did not used BET in this study. However, we will add this method in the introduction part as one of the application method for the porosity measurement with additional references.
We appreciate your comments again.